# A High-Throughput In Vitro Assay for Screening Rice Starch Digestibility

**DOI:** 10.3390/foods8120601

**Published:** 2019-11-21

**Authors:** Michelle R. Toutounji, Vito M. Butardo, Wei Zou, Asgar Farahnaky, Laura Pallas, Prakash Oli, Christopher L. Blanchard

**Affiliations:** 1School of Biomedical Sciences, Charles Sturt University (CSU), Wagga Wagga, NSW 2650, Australia; mtoutounji@csu.edu.au (M.R.T.); wei.zou@csiro.au (W.Z.); asgar.farahnaky@rmit.edu.au (A.F.); 2Australian Research Council (ARC) Industrial Transformation Training Centre (ITTC) for Functional Grains, Graham Centre for Agricultural Innovation, CSU, Wagga Wagga, NSW 2650, Australia; 3Faculty of Science, Engineering and Technology, Swinburne University of Technology, Hawthorn, VIC 3122, Australia; 4Agriculture and Food Innovation Centre, The Commonwealth Scientific and Industrial Research Organisation (CSIRO), Werribee, VIC 3030, Australia; 5School of Science, Royal Melbourne Institute of Technology (RMIT) University, Bundoora West Campus, Melbourne, VIC 3083, Australia; 6NSW Department of Primary Industries (DPI), Yanco Agricultural Institute, Yanco, NSW 2703, Australia; lapallasathena@gmail.com (L.P.); prakash.oli@dpi.nsw.gov.au (P.O.)

**Keywords:** digestibility, high throughput, glycaemic index, starch, rice, screening

## Abstract

The development of rice that can produce slow and steady postprandial glucose in the bloodstream is a response to alarmingly high global rates of obesity and related chronic diseases. However, rice grain quality programs from all over the world currently do not have access to a high-throughput method to distinguish rice breeding materials that are digested slowly. The objective of this study was to develop a high-throughput in vitro assay to screen the digestibility of cooked white rice grains and to investigate its ability to differentiate rice genotypes with a low starch digestibility rate. The digestibility rate and extent of three commercial rice genotypes with diverse GI values (Doongara, Reiziq and Waxy) were successfully differentiated using the protocol. Further investigations with eight rice genotypes indicated the percentage of starch hydrolysed at a single time point of the assay (SH-60) successfully differentiated genotypes with a low digestibility rate (the SH-60 of Doongara and YRL127 was 50% and 59%, respectively) from genotypes with an intermediate or high digestibility rate (SH-60 values were between 64% and 93%). Application of this methodology in rice breeding programs may assist in the screening and development of new varieties with a desirable postprandial glycaemic response.

## 1. Introduction

Rice (*Oryza sativa* L.) is a traditional staple crop, feeding more people over a longer period of time than any other grain [1]. The digestibility of white rice is nutritionally important as this food meets the energy needs of roughly half the world’s population. Primarily composed of starch, white rice generally causes a marked increase in blood glucose levels after enzymatic amylolysis. However, the increasing incidence of obesity and related chronic disease has led to consumer demand for highly satiating rice that can provide slow and steady postprandial glucose in the blood stream. The diversity of genes associated with controlling rice digestibility provides an opportunity to select new varieties with reduced digestibility using classical breeding [2].

Testing the digestibility of rice in vivo (with human subjects) to obtain a measurement of glycaemic response is very time-consuming (i.e., it requires ethical approval and the recruitment of volunteers) and expensive (e.g., remuneration to trained medical personnel and the disposal of clinical waste). Within the food industry, the glycaemic index (GI) is considered the gold standard measurement for carbohydrate-containing foods; however, testing for GI currently costs thousands of dollars per sample. Additionally, agricultural breeding facilities are usually not set up to conduct human clinical trials and cannot afford to test the GI of advanced rice lines (with potential low GI values) every year. A good correlation between in vitro and in vivo methods has already been demonstrated for rice digestibility [3,4,5]. Therefore, there is a need for a simple, reliable and cost-effective in vitro digestion assay that would not necessarily replicate the complex interactions between food and the oral and the gastrointestinal tract, but could identify rice breeding materials with a low rate of starch digestibility.

A number of in vitro methods have been used to analyse the starch digestibility of different types of rice [6,7,8,9], which are usually based on earlier methods developed in the 1990s [10,11]; however, the protocols are inconsistent. Sample preparation is different, with physical destruction of the grains occurring either before cooking, to produce flour, or after cooking (e.g., using a mincer, chopper or homogeniser). There is no exact enzyme recipe; some methods use amylases (often pancreatic alpha-amylase), whereas other assays employ amylases with some combination of proteases, lipases and ribonucleases. Moreover, the current in vitro methods are not high-throughput and therefore are unable to satisfy the quick turn-around time required for breeders to select lines to be carried through to the next round of the breeding cycle. A high-throughput digestibility method must be rapid and have a reasonable cost per assay.

In this study, we aimed to develop and assess a high-throughput in vitro assay to distinguish rice genotypes with a low digestibility rate.

## 2. Materials and Methods

### 2.1. Materials

Magnetic stir bars (Cowie, PTFE-coated, octahedral, 38 × 8 mm) were purchased from Aim Scientific (Prospect, SA, Australia). Sodium acetate anhydrous (CH_3_COONa) and sodium hydroxide pellets (NaOH) were obtained from Chem-Supply Pty Ltd. (Gillman, SA, Australia). Glacial acetic acid (CH_3_COOH) and magnesium chloride anhydrous (MgCl_2_) were sourced from Sigma-Aldrich (Castle Hill, NSW, Australia). Calcium chloride dihydrate (CaCl_2_.2H_2_O) was purchased from Thermo Fisher Scientific (Scoresby, Australia). Alpha-amylase (porcine pancreas, 100,000 U/g), amyloglucosidase (*Aspergillus niger*, 3300 U/mL), and D-Glucose Assay Kit (oxidase/peroxidase, GOPOD format) were sourced from Megazyme International Ireland Ltd. (Wicklow, Leinster, Ireland). Milli Q quality (Millipore, Bedford, MA, USA) water was used for the assay.

Seven rice genotypes were provided in paddy form by the NSW Department of Primary Industries (DPI) Yanco Agricultural Institute (YAI): Doongara, Koshihikari, Opus, Reiziq, Sherpa, Topaz, and YRL127. These genotypes were grown in the Murrumbidgee Irrigation Area (NSW, Australia) and harvested in 2016. One white glutinous rice genotype (waxy rice, Thailand), manufactured in November 2016, was purchased from a local grocery store (Wagga Wagga, NSW, Australia).

### 2.2. Sample Preparation

Australian paddy rice samples were dehulled with the Testing Husker THU 35A (Satake Engineering Co., Ltd., Tokyo, Japan) and polished using the OnePass Rice Whitening & Caking Machine (Satake Engineering Co., Ltd., Tokyo, Japan) at Yanco Agricultural Institute, NSW DPI. White rice grains were stored in sealed plastic specimen jars at 4 °C and then equilibrated to room temperature at least 24 h prior to analysis.

Grain samples were freshly cooked on the day of starch digestion analysis. Water (5 mL) was added to four (preweighed) intact, white rice grains in 150-mL Schott bottles. Bottles were tightly capped and immediately submersed in a vigorously boiling water bath for 30 min to ensure that samples were fully cooked. The sufficiency of cooking was routinely tested by squashing cooked white grains between two glass slides. The absence of a white core was used as a visual indication of well-cooked grains. Freshly cooked samples were transiently stored in a 60 °C water bath, and a digestibility assay was carried out immediately to prevent starch retrogradation. 

### 2.3. Enzyme Optimisation

Optimisation of starch digestibility assays was done in two stages, following the cooking conditions described in Section 2.2. In the first stage, alpha-amylase (AA) or amyloglucosidase (AMG) enzymes were added sequentially, while the concentration of the other enzyme kept constant (Figure 1). Samples were digested with fixed or varying concentrations of AA for 3 h; aliquots were heated at 100 °C for 10 min to inactivate AA, and then further digested with varying or fixed concentration of AMG for 20 min. Another setup was prepared where only varying concentrations of AMG were used. The optimum AA and AMG concentrations were determined by monitoring the kinetic profile of starch digestion with varying concentrations of each enzyme. In the second stage, dual-enzyme assays were conducted using the optimal concentrations of AA (1 U/mL) and AMG (5 U/mL) in the total 50 mL volume, either added simultaneously or sequentially (Figure 1). The effect of the sequential or simultaneous addition of AMG was then tested to determine an optimal and convenient method for screening the starch digestibility of cooked white grains.

### 2.4. In Vitro Starch Digestion Assay

White rice grains were cooked as described in Section 2.2, and 40 mL of sodium acetate buffer (0.2 M, pH 6.0), which had previously been equilibrated to 60 °C, was added to each bottle. All samples were stirred for exactly 5 min at 200 rpm and then equilibrated to 37 °C. At this stage, duplicate aliquots (0.2 mL) were sampled; this was considered the 0 min time point. Five millilitres of a working enzyme solution were added so that the final 50 mL volume was digested with 1 U/mL pancreatic α-amylase and 5 U/mL amyloglucosidase. The mixture was incubated at 37 °C for 3 h with magnetic stirring at 200 rpm, using a submersible stirrer that can stir up to 15 samples at a time (2mag-USA, MIXdrive 15HT Stirring Drive). The temperature was maintained at 37 °C using a recirculating water bath (2mag-USA, MIXbath S Stainless Steel Bath Tank with Julabo, Corio C Immersion circulator). Digesta in duplicate aliquots (0.2 mL) were sampled from each bottle using a 1-mL micropipette and immediately frozen using liquid nitrogen. To monitor the kinetics of starch hydrolysis, the following sampling time points were used: 5, 10, 20, 30, 45, 60, 90, 120, and 180 min. The assay incubation time of 3 h approximately follows the time taken for substrates to transit through the small intestine.

Digesta samples were heated at 100 °C for 10 min to inactivate enzymes and then centrifuged at 13,000 rpm for 10 min. The glucose concentration of the supernatant was measured using a D-Glucose Assay kit (GOPOD method, Megazyme International Ireland, Bray, Ireland) and a FLUOstar^®^ Omega spectrophotometer (BMG Labtech, Ortenberg, Germany). The digestibility of samples was calculated according to the following equation and plotted as a percentage of starch hydrolysed over time:(1)$$\%\;Starch\;hydrolysed\;=\;\frac{weight\;of\;glucose\;in\;supernatant\;\times 0.9}{dry\;weight\;of\;starch\;in\;solution}\;\times \;100,$$
where 0.9 is the molar mass conversion from glucose to anhydroglucose (the starch monomer unit).

### 2.5. Statistical Analysis

Each sample was measured in triplicate, with the glucose concentration analysed in duplicate. All results were reported as means ± standard deviations (SD). All analyses were performed using GraphPad Prism version 7.03 for Windows (GraphPad Software Inc., La Jolla, CA, USA). Enzyme kinetic parameters were obtained by fitting the obtained data to the Michaelis‒Menten equation using nonlinear regression with the least squares fit. To assess statistically significant differences between more than two groups of data, a two-way ANOVA test was applied, with Tukey’s test (*p* < 0.05) used to compare each different group. For SH-60 values, comparison of means is denoted by letters, with similar letters signifying no significant difference using Tukey’s test (*p* < 0.05).

## 3. Results

### 3.1. A High-Throughput In Vitro Assay Proposed Specifically for the Digestibility of Cooked Rice

To allow for rapid screening of samples, modifications were made to the in vitro digestion assay (as described in Section 2.4). Sampling was only taken at a single time point (60 min), and digesta aliquots were immediately heated to a 100 °C to inactivate enzymes, omitting the previous step of snap-freezing with liquid nitrogen. An overview of the assay is presented in Figure 2.

### 3.2. A Single Time Point Measurement of Starch Hydrolysis Is an Effective Method for Rapid Estimation of the Digestibility of Rice Genotypes

The proposed digestion method for cooked rice grains was elucidated by monitoring and comparing the kinetic profile of starch hydrolysed using varying concentrations and simultaneous or sequential addition of AA and AMG (Appendix A). Irrespective of the enzyme combination used, all sets of data displayed monophasic digestion behaviour. For digestion using sequential addition of AA (at various concentrations) and excess AMG, AA at 1 U/mL resulted in starch hydrolysis value at 60 min (SH-60) of 54 ± 2.6%. For dual-enzyme digestibility, sequential addition of 1 U/mL AA and 5 U/mL AMG produced an SH-60 value of 51 ± 1.1%. For digestion with AMG alone, using 10 U/mL gave an SH-60 value of 55 ± 0.49%, but using a very high concentration of AMG to screen thousands of rice lines in a breeding population will be prohibitively expensive.

An assay using sequential addition of 1 U/mL AA and 5 U/mL AMG was compared with digestion with 5 U/mL AMG alone to elucidate the potential synergistic or antagonistic effects of these enzymes (Appendix A). Nonlinear regression analysis revealed a different curve for each dataset. Digestion with AMG alone was slower than digestion with AA/AMG, with SH-60 values of 37 ± 3.8% and 49 ± 0.6%, respectively. The kinetic profile of starch digestion in rice using simultaneous addition of AA and AMG was also assessed and compared with digestion by sequential addition of these enzymes (Figure 3). The SH-60 value for cooked grains digested by AA/AMG added simultaneously (54 ± 6.2%) was not statistically different from the digestion by the enzymes added sequentially (60 ± 10.7%). In the final 50 mL volume, we used the optimal enzyme concentration of 1 U/mL AA and 5 U/mL of AMG added simultaneously (Appendix A) to reduce the analysis time of the assay.

### 3.3. Rice Genotypes with Varying GI Values Can Be Differentiated Using the Digestibility Assay

The digestibility kinetics of three rice genotypes, Doongara, Reiziq and Waxy, were assessed using the in vitro assay to test whether the optimised method was suitable for rice with a wider range in GI scores. The kinetic profiles of starch hydrolysis showed a wide variation in the digestion rate and extent of digestion between the three rice genotypes (Figure 4). The starch digestograms showed the same monophasic behaviour; however, different curves for each dataset resulted upon nonlinear regression analysis. Comparisons between the three rice genotypes showed statistically significant differences (*P* ≤ 0.001) at every digestion time point. Complete hydrolysis of Waxy rice was observed after 90 min, whereas the hydrolysis of Reiziq and Doongara at the final time point of the assay (180 min) reached 90 ± 3.7% and 79 ± 4.8%, respectively. The digestion of the rice genotypes at each time point followed the trend: Waxy > Reiziq > Doongara. In terms of SH-60, Doongara clearly showed a substantially lower value (50 ± 6.0%) compared to Reiziq (73% ± 5.5%) and Waxy (93% ± 2.0%).

### 3.4. The In Vitro Assay Differentiated Eight Rice Genotypes Based on Their Digestibility

Eight rice genotypes were digested in vitro with sampling at the 60 min time point to obtain the corresponding SH-60 values (Figure 5). Doongara was the least digestible rice among the eight genotypes, with a low SH-60 value of 50 ± 6.0% and Waxy was the most digestible, with an SH-60 value 93 ± 2.0%. The remainder of the genotypes, Koshihikari, Opus, Reiziq, Sherpa, Topaz, and YRL127, had SH-60 values between 59% and 93%.

## 4. Discussion

A simple in vitro assay has been described, in which the starch digestibility rate of cooked white rice grain samples can be measured (Figure 2). It could be used in rice breeding programs as a useful phenotyping tool. We do not claim that this assay will produce values that accurately predict the glycaemic behaviour of rice, but it does enable the rapid identification of rice genotypes with a low starch digestibility rate.

As the proposed protocol was specifically developed for rice, the sample preparation (cooking and destruction of the food structure) was given great consideration. Rice is always cooked prior to consumption. Thus, cooked rice was freshly prepared immediately prior to analysis and kept warm at 60 °C until the buffer was added. All rice varieties in this protocol were cooked to completion using an extended cooking time [12] to overcome the potential effect of variations in cooking time. In addition, we used intact (unbroken) grains prior to cooking. While the use of rice flour in an in vitro digestion assay would more easily fit into rice breeding programs, which already incorporate flour samples for other analytical tests, cooked grains were used to reflect the way rice is normally consumed. The number of grains described in our assay was based upon the use of 50 mg available carbohydrates adopted in other in vitro assays [11,13]. A small amount of sample per test was also justified for the intended application, as there is often a limited amount of breeding material available. The same vessels were used during cooking and digestion of samples to prevent sample loss and improve the control of sample temperature.

As with most in vitro digestion models reported in the literature, the assay reported here uses a static system (maintaining constant substrate-to-enzyme ratios, salt, etc.) maintained at 37 °C, allowing for simplicity and ease of sampling [14]. Continuous magnetic stirring was used during digestion to achieve homogenous mixing [14]. Preliminary investigations using a commercially available semi-automated in vitro digestibility instrument to rank the digestibility of rice varieties (Appendix A) were found to have limited applicability in the rice breeding program due to the longer running time (5 h) and higher capital outlay and operational running costs. Also, this commercial method was not found to be suitable as a high-throughput assay due to the limited number of samples that could be analysed in one day. Hence, a custom screening method that quickly scored the digestibility of white rice grains was developed for deployment in rice breeding programs.

A high-throughput in vitro method needs to simulate rice starch enzymatic digestion at the best rate, with some compromises made to increase the reliability and decrease the cost of the assay. Pancreatin (a mixture of proteases, amylases, lipases and ribonucleases) was not included in our assay as enzyme activities of commercial preparations of pancreatin differ by source and grade, introducing batch-to-batch variation and a requirement to recalibrate the assay regularly [15]. There are also some biosafety and regulatory hurdles regarding importation to certain countries, such as Australia. Simulated oral phase and gastric phases were also excluded from the in vitro digestion. Woolnough et al. [16] reported that the oral phase by salivary α-amylase is not necessary when chewing is simulated (in our case, stirring for 5 min). In addition, it was reported that the hydrolysis of cooked rice using a simulated gastric phase (using pepsin in a high-pH environment) prior to simulated intestinal phases (with AA and AMG) was not significantly different from samples only hydrolysed by simulated intestinal digestion with AA and AMG [15]. Hence, after a series of optimisation steps, the focus of this rapid assay was starch digestibility with the simultaneous addition of AA and AMG.

Doongara was used in enzyme optimisation assays due to the available in vivo clinical data, characterising it as a slowly digestible rice variety [17,18,19,20]. One study reported that the GI of Doongara, 64 ± 9, was significantly lower than other varieties (Calrose, Pelde and Waxy), which ranged from 83 ± 13 to 93 ± 11 [17]. In the international table of GI and GL values by Foster-Powell et al. [18], Doongara was classified as having a low glycaemic index (GI), i.e., 55 or less. Similarly, Williams et al. [19] reported Doongara as having a low GI value (51 ± 6), compared to Basmati (59 ± 6) and other varieties (Amaroo, Opus, Kyeema, Langi, and Koshihikari), which ranged from 61 ± 8 to 89 ± 8. More recently, ethnic differences in postprandial glycaemia were reported whereby Doongara was low GI (reported average of 55, ranging from 48 to 63) for European volunteers and intermediate GI (reported average of 67, ranging from 58–76) for Chinese volunteers [20].

During assay development, the digestibility of cooked Doongara rice grains was measured at a wide range of AA and AMG concentrations. To ensure the complete hydrolysis of starch (when digestion assessment is by glucose analysis), excess AMG is needed to convert 100% of AA reaction products to glucose. Using excess AMG is important because the measurement is simplified by completely converting the intermediate products of AA activity (i.e., maltooligosaccharides) into glucose. Increasing the enzyme concentration of AMG at a fixed AA activity was shown to increase the rate of in vitro digestion (Appendix A). The optimal enzyme concentrations for the digestibility of Doongara to produce an SH-60 value around 55%, roughly following the GI ranking system of 0–100, was found to be 1 U/mL AA and 5 U/mL AMG in the final volume of 50 mL.

The potential synergistic and antagonistic effects of AA and AMG were investigated within the system (sample preparation and equipment setup) of our proposed assay. There is a need to determine the optimal synergistic concentration for AA and AMG because the former cleaves the glucan chain internally (endo-enzyme), while the latter releases the glucose molecule from the external reducing ends (exo-enzyme) [21]. The action of AA and AMG on native starch granules has been observed to be synergistic via two mechanisms: 1) the AA randomly splits the substrate molecules on the granular surface, providing new nonreducing end groups to AMG; and 2) AMG can “peel” starch molecules from the surface of a granule, exposing newly nonreducing end groups for attack by AA [22]. Evidence of the synergism between AA and AMG activity for raw starch granules has been demonstrated by experiments where the released glucose value is more than twice that observed in the mixed-enzyme system compared to the corresponding value for the digestion by AMG alone [22,23]. A similar trend was observed here for cooked rice grains, albeit to a much lesser degree. The digestion curve of AMG alone displays a slower rate compared to AMG digestion after pretreatment with AA (Appendix A). This suggests that AA is a rate-limiting enzyme during starch digestion for cooked rice grains, at least at the concentrations we used. These findings are contradictory to the antagonistic action of AA and AMG on cooked maize and potato starch [23]. However, it must be noted that the samples included in the Zhang et al. [23] study were of different botanical origin (maize and potato) and, perhaps more importantly, underwent thorough processing as a commercially isolated starch product. The significance of food structure in influencing amylolytic enzyme activity has been reported previously [24,25,26]. In particular, cell wall encapsulation can influence starch digestibility through limited access to digestive enzymes and/or substrate and product release [27]. When in vitro digestibility experiments are used to predict the postprandial glycaemic response of foods, close attention must be paid to sample preparation to ensure that the food structure closely mimics real-life conditions. Thus, we reiterate the importance of using rice grains, rather than starch or flour, in our proposed method.

In our optimization experiments, when both AA and AMG were used to digest cooked rice grains, there was no significant difference in the digestion rate when enzymes were added simultaneously compared to sequential addition (Figure 3). Based on this result, simultaneous addition of AA and AMG was used in the final proposed assay to increase the ease and convenience of the assay. The kinetic starch hydrolysis profiles of three commercial rice genotypes (Doongara, Reiziq and Waxy) were successfully differentiated using the 3 h version of the protocol (Figure 4), with Waxy having the fastest rate and Doongara the slowest. Doongara has been demonstrated to have a significantly lower GI compared to other rice varieties [19], which may be attributed to its high amylose content, as shown by previous studies [4,17,28]. In order to be suitable for high-throughput screening, the assay needs to be rapid, scalable and inexpensive. We observed that the proportion of starch hydrolysed at 60 min (SH-60) during the in vitro digestion assay could be used as a proxy measure for rice digestibility, significantly reducing the duration of the assay. In our study, eight rice genotypes were compared according to their SH-60 values. Doongara and YRL127 were successfully differentiated from other genotypes as having a low digestibility rate (SH-60 around 55%), with values of 50 ± 6% and 59 ± 3%, (Figure 5). The remainder of the samples had intermediate to high SH-60 values. The SH-60 values of the genotypes seemed to correspond to their intrinsic starch properties (see the Appendix A), which is in agreement with the literature [29].

The proposed in vitro assay allows up to 15 samples to be analysed simultaneously, allowing 60 samples to be easily analysed per day (assuming that 15 samples can be prepared every 2 h). For future optimization of the assay, improved throughput could be achieved with the use of a more rapid time point of the digestion (less than 60 min), provided that the enzyme concentration is limited. However, a major limitation of this research is that the proposed method has not been properly validated on a large sample set. A future experimental design that includes a large number (hundreds or more) of rice genotypes digested in vitro using the proposed assay, with a representative number measured for their glycaemic response in human feeding trials, would provide a more robust approach for estimating the potential postprandial glycaemic response of rice.

## 5. Conclusions

A high-throughput in vitro starch digestibility assay was developed specifically for cooked rice grains. This methodology uses glucose released at a single time point, expressed as hydrolysed starch (SH-60), and can distinguish genotypes with a low digestibility rate. As the digestion model was designed for foods with a very similar composition (i.e., rice genotypes), the comparisons made between samples are inherently more accurate. The main advantages of the proposed assay over current in vitro digestion methods is that it is simple to perform, rapid and relatively inexpensive. The application of this methodology in rice breeding programs offers a practical screening tool for the development of new varieties with a desirable postprandial glycaemic response.

## Figures and Tables

**Figure 1 foods-08-00601-f001:**
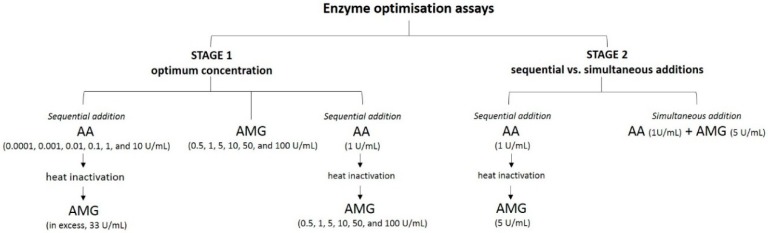
Experimental design of the enzyme optimisation assays using alpha-amylase (AA) and amyloglucosidase (AMG).

**Figure 2 foods-08-00601-f002:**
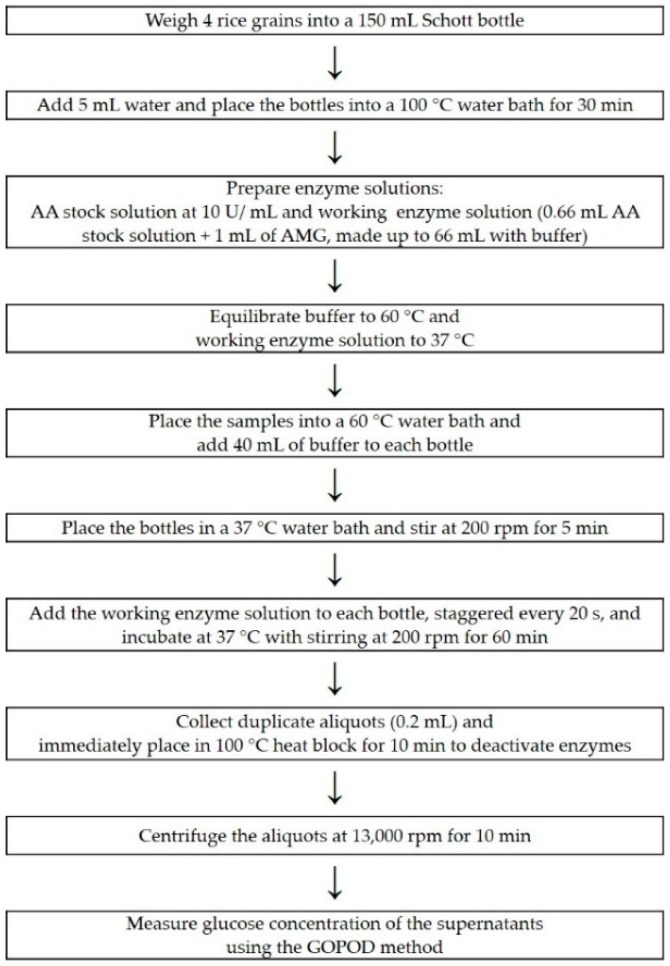
A flow diagram of the high-throughput digestibility assay, with a maximum of 15 samples per assay. Original enzyme activities of alpha-amylase (AA) and amyloglucosidase (AMG) were 100,000 U/g and 3300 U/mL, respectively. The buffer used throughout the assay is 200 mM sodium acetate buffer at pH 6.0.

**Figure 3 foods-08-00601-f003:**
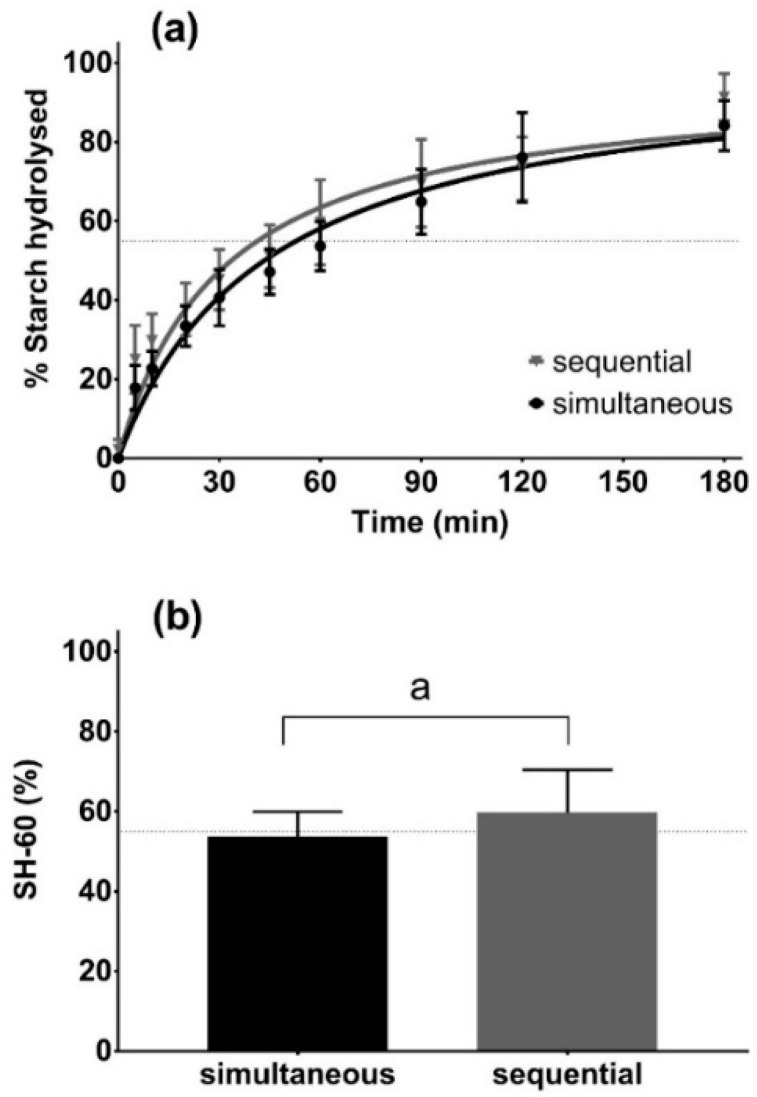
Starch digestogram (**a**) and starch hydrolysed at 60 min (**b**) of cooked Doongara grains digested by 1 U/mL pancreatic α-amylase and 5 U/mL amyloglucosidase, by sequential or simultaneous addition (*n* = 3). Horizontal broken lines signify the SH-60 value of 55%.

**Figure 4 foods-08-00601-f004:**
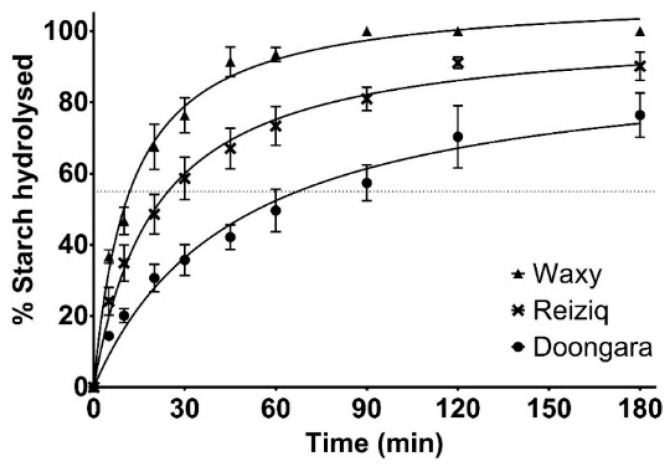
Starch digestogram of three rice genotypes (*n* = 3). Starch hydrolysed at the 60 min time point is highlighted in yellow, and the dotted line denotes that 55% of the starch was hydrolysed.

**Figure 5 foods-08-00601-f005:**
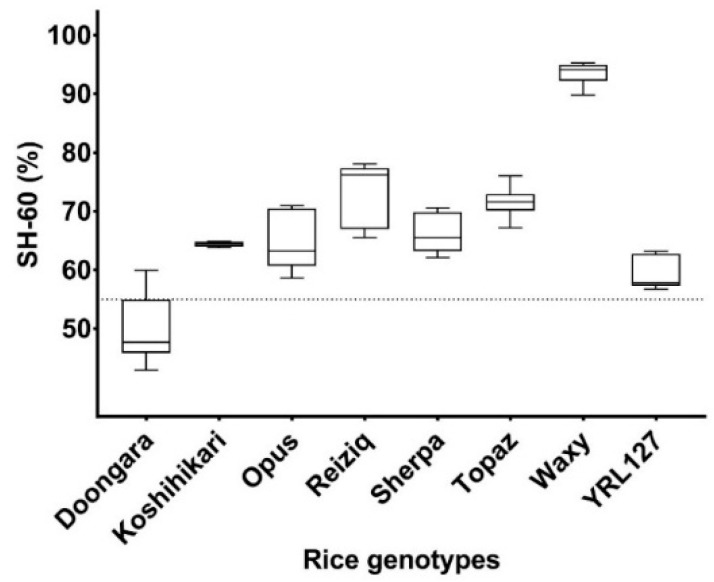
Box and whisker plot of starch hydrolysed at the 60 min time point (SH-60) for eight different rice genotypes (*n* = 3). The dotted line denotes that 55% of the starch was hydrolysed.

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
