# Peer review of "A High-Throughput In Vitro Assay for Screening Rice Starch Digestibility"

_foods, 2019, doi:10.3390/foods8120601_

Round 1

Reviewer 1 Report

This article describes the method development of starch digestibility of various rice cultivators for the screening purpose. I do see clear research objective of this article but do not see the methodology appropriate for the research purpose.

Major concerns-

In this study, single time point of enzymatic hydrolysis as well as combination effects of two different amylolytic enzymes would be proper to differentiate digestibility level of rice starch depending on wide range of varieties. This methodology only displayed a certain condition of enzyme hydrolysis and determined a time point for their own way. Thus, it could not be any standard method for evaluating in vitro digestibility level of various starch source at all. Also, the number of samples were very limited and not enough to generate the acceptable results. The catalytic properties of both alpha-amylase and amyloglucosidase and even the combination effect of them have been well known. Thus, I do not see any significant original findings in this study.

Reviewer 2 Report

General content:

Development of in vitro assay for a rapid screening of low digestibility phenotypes of cooked white rice grains in order to develop low glycaemic index rice cultivars. The main scope was to investigate the ability of the assay to differentiate rice genotypes with varying GI values.

General opinions:

The experimental design seems to be good but the correlation between in vitro results and clinical GI of rice varieties seems to weak and approximate. It is necessary to detail the statistical difference between the in vitro values obtained within and between the rice varieties. However, the proposed method for screening rice varieties should be explained in detail.

MAJOR:

- abstract:

Line 26: it is necessary to include the GI values of rice varieties (Doongara, Reiziq, Waxy) and specify how have been obtained these values

Line 26: it is necessary to specify on the basis of what value the rice cultivars have been differentiated

- Materials and methods:

Line 145: it is necessary to explain better the GI of selected rice Doongara: Which studies show that this rice variety has a low glycaemic index? However, is necessary to insert the references of the clinical data

Line 176: On the basis of what has been deducted as ‘with low starch hydrolysed’ a value of SH-60?

- Results

Line from 194 to 209: All significance values shall be entered and the statistical analysis conducted shall be specified. However, is necessary to detail the type of analysis conducted: within the same variety and between the specified values studied or within the values studied and between the varieties?

Line 237: it is necessary to better explain the GI value of 53 of Doongara rice: how was that value obtained?

Line 246: is necessary to statistically explain better the variation in the digestion rate between the three rice genotypes

Line  253: the value of SH-60 of Waxy rice does not match the one shown in the table 3

Line from 270 to 272: it is necessary to specify how have been obtained the GI values of rice varieties and how the average values was obtained and used. However, is necessary to specify the variation in terms of SD or SEM of average values calculated.

Line from 273 to 276: the correlation between in vitro results and clinical GI of rice varieties seems to weak, unclear and approximate.

Line 278: the values of SH-60 of waxy rice shown in table 3 doesn’t correspond with the values shown in line 253, 262 and 384

MINOR:

- abstract:

Line 28: it is necessary to specify the values of SH-60 of rice varieties

Round 2

Reviewer 2 Report

The manuscript has been reviewed by authors, by taking into account referees' queries.

Minors:
- Materials and methods: 
line 202: specify the types of rice used to perform the in vitro protocol
line 492: as for the Doongara variety, specify the IG values in the literature for the other varieties and insert the appropriate bibliographic references in order to give much more value to the in vitro digested starch data obtained 
